cognition, evolution, behaviour

mate preferences, sex ratio, sex differences, cross-cultural, mating market

**Author for correspondence:**
Kathryn V. Walter
e-mail: kwalter@ucsb.edu

# Sex differences in human mate preferences vary across sex ratios

Kathryn V. Walter[1], Daniel Conroy-Beam[1], David M. Buss[2], Kelly Asao[3], Agnieszka Sorokowska[4,5], Piotr Sorokowski[4], Toivo Aavik[6], Grace Akello[7], Mohammad Madallh Alhabahba[8], Charlotte Alm[9], Naumana Amjad[10], Afifa Anjum[11], Chiemezie S. Atama[12], Derya Atamtürk Duyar[14], Richard Ayebare[15], Carlota Batres[16], Mons Bendixen[17], Aicha Bensafia[18], Boris Bizumic[20], Mahmoud Boussena[19], Marina Butovskaya[21,22], Seda Can[23], Katarzyna Cantarero[24], Antonin Carrier[25], Hakan Cetinkaya[27], Ilona Croy[28], Rosa María Cueto[29], Marcin Czub[3], Daria Dronova[21], Seda Dural[23], Izzet Duyar[14], Berna Ertugrul[14], Agustín Espinosa[29], Ignacio Estevan[30], Carla Sofia Esteves[31], Luxi Fang[32], Tomasz Frackowiak[4], Jorge Contreras Garduño[33], Karina Ugalde González[34], Farida Guemaz[35], Petra Gyuris[36], Mária Halamová[37], Iskra Herak[26], Marina Horvat[38], Ivana Hromatko[39], Chin-Ming Hui[33], Jas Laile Jaafar[40], Feng Jiang[41], Konstantinos Kafetsios[42], Tina Kavčič[43], Leif Edward Ottesen Kennair[17], Nicolas Kervyn[26], Truong Thi Khanh Ha[45], Imran Ahmed Khilji[46], Nils C. Köbis[47], Hoang Moc Lan[45], András Láng[36], Georgina R. Lennard[20], Ernesto León[29], Torun Lindholm[9], Trinh Thi Linh[45], Giulia Lopez[48], Nguyen Van Luot[45], Alvaro Mailhos[30], Zoi Manesi[49], Rocio Martinez[50], Sarah L. McKerchar[20], Norbert Meskó[36], Girishwar Misra[51], Conal Monaghan[20], Emanuel C. Mora[52], Alba Moya-Garófano[46], Bojan Musil[38], Jean Carlos Natividade[53], Agnieszka Niemczyk[4], George Nizharadze[54], Elisabeth Oberzaucher[55], Anna Oleszkiewicz[4,5], Mohd Sofian Omar-Fauzee[56], Ike E. Onyishi[13], Baris Özener[14], Ariela Francesca Pagani[48], Vilmante Pakalniskiene[57], Miriam Parise[48], Farid Pazhoohi[58], Annette Pisanski[52], Katarzyna Pisanski[4,59,60], Edna Ponciano[61,62], Camelia Popa[63], Pavol Prokop[64,65], Muhammad Rizwan[66], Mario Sainz[67], Svjetlana Salkičević[39], Ruta Sargautyte[57], Ivan Sarmány-Schuller[68], Susanne Schmehl[55], Shivantika Sharad[69], Razi Sultan Siddiqui[70], Franco Simonetti[71], Stanislava Yordanova Stoyanova[72], Meri Tadinac[39], Marco Antonio Correa Varella[73], Christin-Melanie Vauclair[74], Luis Diego Vega[35], Dwi Ajeng Widarini[75], Gyesook Yoo[76], Marta Marta Zaťková[35] and Maja Zupančič[44]

[1]Department of Psychological and Brain Sciences, University of California, Santa Barbara, CA 93106, USA
[2]Department of Psychology, University of Texas at Austin, Austin, TX 78712, USA
[3]Department of Psychology, Westminster College, Salt Lake City, UT 84105, USA
[4]Institute of Psychology, University of Wroclaw, Wroclaw 50137, Poland
[5]Smell and Taste Clinic, Department of Otorhinolaryngology, TU Dresden, Dresden 01307, Germany
[6]Institute of Psychology, University of Tartu, Tartu 50090, Estonia
[7]Department of Mental Health, Faculty of Medicine, Gulu University, Gulu 166, Uganda
[8]English Language Department, Middle East University, Amman 11181, Jordan

[9]Department of Psychology, Stockholm University, Stockholm 10691, Sweden

[10]Department of Applied Psychology, NUR International University, Lahore, Pakistan

[11]Institute of Applied Psychology, University of the Punjab, Lahore 54590, Pakistan

[12]Department of Sociology and Anthropology, and [13]Department of Psychology, University of Nigeria, Nsukka 410002, Nigeria

[14]Deparment of Anthropology, Istanbul University, Istanbul 34452, Turkey

[15]North Star Alliance, Kampala, Uganda

[16]Department of Psychology, Franklin and Marshall College, Lancaster 17603, USA

[17]Department of Psychology, Norwegian University of Science and Technology (NTNU), 7491 Trondheim, Norway

[18]Laboratory Education-Formation-Travail (EFORT), Department of Sociology, and [19]Laboratory EFORT, Department of Psychology and Educational Sciences, University of Algiers 2, Algiers 16000, Algeria

[20]Research School of Psychology, Australian National University, Canberra 2601, Australia

[21]Institute of Ethnology and Anthropology, Russian Academy of Sciences, Moscow 119991, Russia

[22]Center for Social Anthropology, Russian State University for the Humanities, Moscow 119991, Russia

[23]Department of Psychology, Izmir University of Economics, Izmir 35300, Turkey

[24]Social Behavior Research Center, Faculty in Wroclaw, SWPS University of Social Sciences and Humanities, Wroclaw 53238, Poland

[25]Psychology Faculty (Center for the Study of Social Behavior), and [26]Louvain Research Institute in Management and Organisations (LOURiM), Université Catholique de Louvain, Louvain-la-Neuve 1348, Belgium

[27]Department of Psychology, Ankara University, Ankara 6560, Turkey

[28]Department of Psychotherapy and Psychosomatic Medicine, TU Dresden, Dresden 1069, Germany

[29]Grupo de Psicología Política y Social (GPPS), Departamento de Psicología, Pontificia Universidad Católica del Perú, Lima 15088, Perú

[30]Facultad de Psicología, Universidad de la República, Motevideo 11200, Uruguay

[31]Universidade Católica Portuguesa, Católica Lisbon School of Business and Economics, Católica Lisbon Research Unit in Business and Economics, Portugal

[32]Department of Psychology, Chinese University of Hong Kong, Hong Kong, People's Republic of China

[33]Escuela Nacional de Estudios Superiores, Unidad Morelia UNAM, Morelia 58190, Mexico

[34]Psychology Department, Universidad Latina de Costa Rica, San José 11501, Costa Rica

[35]Department of Psychology and Educational Sciences, University of Sétif2, Sétif 16000, Algeria

[36]Institute of Psychology, University of Pécs, Pécs 7624, Hungary

[37]Faculty of Social Sciences and Health Care, Department of Psychological Sciences, Constantine the Philosopher University in Nitra, Nitra 94974, Slovakia

[38]Faculty of Arts, Department of Psychology, University of Maribor, Maribor 2000, Slovenia

[39]Department of Psychology, Faculty for Humanities and Social Sciences, University of Zagreb, Zagreb 10000, Croatia

[40]Department of Educational Psychology and Counseling, University of Malaya, Kuala Lumpur 50603, Malaysia

[41]Organization and Human Resource Management, Central University of Finance and Economics, Beijing 102202, People's Republic of China

[42]School of Fine Arts, Aristotle University of Thessaloniki, Greece and Katedra Psychologie, Palacký University Olomouc, Czech Republic

[43]Faculty of Health Sciences, and [44]Department of Psychology, Faculty of Arts, University of Ljubljana, Ljubljana 1000, Slovenia

[45]Department of Psychology, University of Social Sciences and Humanities, Vietnam National University, Hanoi 100000, Vietnam

[46]Department of Psychology, Islamabad Model College for Boys, F-10/4, Islamabad 44000, Pakistan

[47]Center for Research in Experimental Economics and Political Decision Making, Department of Economics, University of Amsterdam, Amsterdam 1081, The Netherlands

[48]Department of Psychology, Università Cattolica del Sacro Cuore, Milan 20123, Italy

[49]Department of Experimental and Applied Psychology, Vrije Universiteit Amsterdam, Amsterdam 1081, The Netherlands

[50]Department of Social Psychology, University of Granada, Grenada 18010, Spain

[51]Department of Psychology, University of Delhi, Delhi 110021, India

[52]Department of Animal and Human Biology, Faculty of Biology, University of Havana, Havana, Cuba

[53]Department of Psychology, Pontifical Catholic University of Rio de Janeiro, Rio de Janeiro 22451-000, Brazil

[54]Department of Social Sciences, Free University of Tbilisi, Tbilisi 2, Georgia

[55]Faculty of Life Sciences, University of Vienna, Vienna 1090, Austria

[56]School of Education, Universiti Utara Malaysia, Sintok 6010, Malaysia

[57]Institute of Psychology, Vilnius University, Vilnius 1513, Lithuania

[58]Department of Psychology, University of British Columbia, Vancouver, Canada V6T 1Z4

[59]Equipe de Neuro-Ethologie Sensorielle (ENES), Centre de Recherche en Neurosciences de Lyon (CRNL), Centre National de la Recherche Scientifique (CNRS), Jean Monnet University, Saint-Etienne, France

[60]CNRS National Center for Scientific Research, Dynamic Language Laboratory, University Lyon 2, Lyon, France

[61]Institute of Psychology, University of the State of Rio de Janeiro, Rio de Janeiro 21941-901, Brazil

[62]Center of Social Studies, University of Coimbra, 3004 Coimba, Portugal

[63]Department of Psychology—Institute of Philosophy and Psychology 'C. Rădulescu Motru' of Romanian Academy, UNATC Bucharest, Bucharest, Romania

[64]Department of Environmental Ecology and Landscape Management, Faculty of Natural Sciences, Comenius University, Bratislava 84215, Slovakia

[65]Institute of Zoology, Slovak Academy of Sciences, Bratislava 84506, Slovakia

[66]Department of Psychology, University of Haripur, 22620, Pakistan

[67]Escuela de Psicología, Pontificia Universidad Católica de Chile, Santiago 8331150, Chile

[68]Center for Social and Psychological Sciences, Institute of Experimental Psychology SAS, Bratislava, 84104, Slovakia

[69]Department of Applied Psychology, Vivekananda College, University of Delhi, Delhi 110095, India

[70]Department of Management Sciences, DHA Suffa University, Karachi 75500, Pakistan

[71]School of Psychology, Pontificia Universidad Catolica de Chile, Santiago 8331150, Chile

[72]Department of Psychology, South-West University 'Neofit Rilski', Blagoevgrad 2700, Bulgaria

[73]Department of Experimental Psychology, Institute of Psychology, University of São Paulo, São Paulo 05508-030, Brazil

[74]Instituto Universitário de Lisboa (ISCTE-IUL), CIS-IUL, Lisboa 1649-026, Portugal

[75]Fakultas Ilmu Komunikasi, Universitas Prof. Dr Moestopo (Beragama), Jakarta 10270, Indonesia

[76]Department of Child and Family Studies, Kyung Hee University, Seoul 024-47, Republic of Korea

(iD) KVW, 0000-0002-4574-2900; PS, 0000-0001-9225-9965; CB, 0000-0002-3833-7667; IE, 0000-0003-4743-1310; CSE, 0000-0002-7307-8373

A wide range of literature connects sex ratio and mating behaviours in non-human animals. However, research examining sex ratio and human mating is limited in scope. Prior work has examined the relationship between sex ratio and desire for short-term, uncommitted mating as well as outcomes such as marriage and divorce rates. Less empirical attention has been directed towards the relationship between sex ratio and mate preferences, despite the importance of mate preferences in the human mating literature. To address this gap, we examined sex ratio's relationship to the variation in preferences for attractiveness, resources, kindness, intelligence and health in a long-term mate across 45 countries ($n = 14\,487$). We predicted that mate preferences would vary according to relative power of choice on the mating market, with increased power derived from having relatively few competitors and numerous potential mates. We found that each sex tended to report more demanding preferences for attractiveness and resources where the opposite sex was abundant, compared to where the opposite sex was

scarce. This pattern dovetails with those found for mating strategies in humans and mate preferences across species, highlighting the importance of sex ratio for understanding variation in human mate preferences.

## 1. Introduction

The relationship between sex ratio and reproductive processes has been studied across species and mating behaviours [1,2]. For example, shorebird mating systems tend to vary across sex ratios: females tend to have multiple mates in species with typically male-biased sex ratios, whereas males tend to have multiple mates in species with female-biased sex ratios [3]. Additionally, fluctuations in sex ratio within a single species can also be associated with variance in mating behaviours. For instance, the male European bitterling, a freshwater fish, changes mating tactics from defending territory to direct competition as the number of same sex rivals increases [4]; female honey locust beetles increase competitive mating effort as females become more abundant than males [5]; and female guppies display stronger preferences for orange-coloured males as males outnumber females [6].

Yet, despite the breadth of research on sex ratio and mating in non-human animals, research on sex ratio and human mating is surprisingly narrow. Within this literature, most work has examined sex ratio's relationship to 'mating strategy' [7]—one's investment in long-term, committed mating, as opposed to short-term, uncommitted mating— and its consequences (e.g. for marriage rates [8]). Despite mate preferences being among the most important topics in the human mating literature [9], comparatively little empirical attention has been given to the relationship between sex ratio and mate preferences. Here, to address this gap in the literature, we examined the relationship between sex ratio and mate preferences in a large cross-cultural sample spanning 45 countries around the world, and find evidence that mate preferences vary systematically with the ratio of potential mates to potential competitors.

Human males and females face a key challenge of finding and attracting long-term mates that are both desirable and available. An imbalanced sex ratio, where the number of males and the number females in a population are unequal, exacerbates this challenge by affecting the supply and demand of mating opportunities [10] (cf. [11]). The more abundant sex has a reduced probability of gaining access to potential partners, whereas the scarcer sex has access to a wider array of potential partners. The consequences of sex ratio imbalance are made worse by the fact that human mating systems tend to be marked by relative monogamy and mutual mate choice [12,13]. Therefore, power on the mating market—power to express and fulfil one's desires— lies with the sex in demand: the scarcer sex. Throughout human evolutionary history, individuals endowed with a mating psychology sensitive to these power differentials, able to upregulate the expression of sex-typical desires when one's sex is scarce and downregulate these desires when one's sex is numerous, would probably have had a competitive advantage over individuals with desires that remained static in the face of shifting contexts.

The effects of this sex differential in market power in humans have primarily been studied in the context of mating strategy attitudes and behaviours. Men, owing to their smaller obligatory investment in offspring, can potentially derive greater direct fitness benefits from acquiring multiple mates than can women [14]. Consequently, across cultures, men on average report greater willingness to engage in sex without commitment—a higher 'sociosexuality'—than women [15,16].

However, this average sex difference is qualified by the finding that nation-level indices of sociosexuality are higher in countries where men are scarce, and therefore have more market power [15]. This replicates outside of industrialized cultures: for instance, one study found that men's sociosexuality varied across communities within the indigenous Makushi as a function of the sex ratio of those communities [17]. Behaviourally, marriage rates increase and divorce rates decrease when women are scarce [18–20].

The same market forces that shape sociosexuality should also have consequences for mate preferences. Mate preferences, in general, have received extensive empirical attention. A large body of the literature has documented universal trends in long-term mate preferences, including the importance of kindness, intelligence and health, and universal sex differences in preference for physical attractiveness, resources and relative age [21,22]. Importantly, these preferences do predict real mate choices [23–26] (cf. [27]). While these average patterns of mate preferences have been consistently documented across time and cultures, the effect sizes of sex differences in mate preferences do vary across cultures. Sex ratio may be a source of the cross-cultural variation in mate preferences, just as it is for mating strategy.

The limited existing literature examining sex ratio and human mate preferences is marked by inconsistencies. One large cross-cultural study found that both men and women placed greater importance on good financial prospects, refinement and neatness, and other qualities in countries where men were more numerous than women [28]. This is unexpected from a market economic perspective, where the change in men's and women's preferences should be inversely related due to differing relative power on the mating market. Yet this cross-cultural study, while impressive in sample size, had important methodological limitations, including analysing exclusively aggregate country-level correlations and incorporating a measure of preferences that allowed only limited variation [29]. Another study found that in Canadian cities where women were relatively scarce, they placed more emphasis on the physical attractiveness of potential mates in newspaper ads [30]; however, this study did not examine men's preferences. Lastly, measurement of sex ratio is not consistent across prior studies examining the consequences of sex ratio in humans, limiting generalizations across findings. For instance, studies vary in how they define sex ratio, and whether operational sex ratio (only individuals able to reproduce) or adult sex ratio (all individuals considered adults, including elderly) is the key variable. Therefore, studies vary in the age range for which sex ratio is estimated, with ranges including ages 15–49 [28], ages 18–45 [17], ages 20–50 [20], ages 16–39 [31] and ages 15–64 [29]. Addressing these limitations will be important for understanding how human mate preferences relate to the scarcity or abundance of potential mates in their environment and whether this relationship is consistent with prior psychological, anthropological and biological literature.

In the present investigation, we examined the relationship between mate preferences and sex ratio in a large, 45-country sample. First, we asked both men and women about their preferences for five traits in an ideal long-term mate and examined how these preferences varied across countries as a function of sex ratio. We analysed the data using multilevel models to account for the nested nature of the data and to take advantage of the large sample size, rather than relying on aggregate correlations. Furthermore, in an attempt to correct for issues in prior work, we incorporated city-level sex ratio and multiple measures of sex ratio at the country-level. Additionally, we measured preferences both in an absolute form (the trait value indicated as ideal in a potential mate) and as a relative preference (ideal trait value relative to the trait distribution available in each country) to allow for clearer comparisons across samples.

Overall, both men and women were predicted to have greater absolute and relative preferences where they were the scarcer sex. Members of the more numerous sex were predicted to have the opposite pattern and express less demanding mate preferences.

## 2. Method

### (a) Participants

Data were collected in 2016, from $n = 14\,487$ (7961 female, 54.95%) participants in 45 countries. All participant data were collected in person because online samples tend to be less representative of populations in developing countries [32]. Each study site collected data from both university populations and community samples. However, due to a lack of records from about half of the sites, there is incomplete information about the percentage of each type of sample. From the sites that did keep records ($n = 6637$), 47.14% ($n = 3129$) came from community samples. Age of participants ranged from 18 to 91 years old ($Mdn = 25$, $M = 28.79$, s.d. = 10.64). Of the total sample, most participants reported being in ongoing, committed relationships ($n = 9236$, 63.75%). Overall, participants tended to be from large cities, to be well-educated, and have average economic situations (detailed city and participant demographic information is in the electronic supplementary material).

Surveys were translated if necessary and distributed to participants through a collaborative cross-cultural data collection project. For more details and a complete list of countries and sample sizes, see the electronic supplementary material.

The data from this cross-cultural data collection process have been used in other papers published previously [22,33–35].

### (b) Measures

#### (i) Mate preferences and participant traits

Participants completed a 5-item questionnaire on ideal mate preferences for a long-term romantic partner. Participants rated their ideal romantic partner on five traits: kindness, intelligence, health, physical attractiveness and good financial prospects. All items were rated on bipolar adjective scales ranging from 1 (very unintelligent; very unkind; very unhealthy; very physically unattractive; very poor financial prospects) to 7 (very intelligent; very kind; very healthy, very physically attractive; very good financial prospects). Using the same scales as for preferences, participants additionally rated themselves on the same five traits: kindness, intelligence, health, physical attractiveness and good financial prospects. We also asked participants about their sex (male/female).

#### (ii) Sex ratio

We used a variety of measures of sex ratio from publicly available databases. As there is no literature standard measure of sex ratio in humans, we wanted to examine the relationship between mate preferences and a variety of measures of sex ratio. We used country-level sex ratio at birth [36], adult sex ratio (ages 18+) [37], sex ratio for ages 15–49 [37], sex ratio for ages 15–64 [37] and city-level overall sex ratio [38–42]. For city-level sex ratio, we cross-checked local sources of information about sex ratio when possible. We also confirmed that city-level sex ratio was correlated with country-level sex ratio measures (sex ratio at birth, $r = 0.16$; adult sex ratio, $r = 0.79$; sex ratio ages 15–49, $r = 0.33$; sex ratio ages 15–64, $r = 0.57$). To explore whether participant's mate preferences were influenced by the sex ratio of their own age group, we also examined the relationship between mate preferences and sex ratio of narrower age categories: sex ratio ages 15–24, sex ratio ages 25–49 and sex ratio 50+ [37] (see electronic supplementary material). For every sex ratio measure, we attempted to collect the publicly available data that were closest to 2016, which was the year we collected preferences and traits from participants.

#### (iii) Control variables

Each analysis was conducted twice; first without controls, and then with all control variables simultaneously. Control variables include latitude [43], world region (defined in [44]), country religion [45], GDP per capita [46], gender equality (a composite measure of gender equality from a principal component analysis of three measures of gender equality: the Global Gender Gap Index [47], the Gender Inequality Index (GII) [48] and the Gender Development Index [49]), income inequality (the Gini Index [50]) and socioeconomic development (socioeconomic development is defined [51] as the summed standardized scores for country's gross national income (GNI) [52], infant survival rate [53], life expectancy [54] and the percentage of population that is urban [55]). For all controls, we attempted to collect the publicly available data that were closest to 2016, which was the year we collected preferences and traits from participants (see electronic supplementary material for more details and justification of the control variables).

### (c) Analyses

We conducted all primary analyses using multilevel models. The general format of these models predicted preference variables as the outcome variable using the interaction of sex and sex ratio variables; participants were nested within countries or cities, as appropriate. The models included random effects for both slopes and intercepts. Multilevel models provide advantages over traditional approaches for analysing these kinds of cross-cultural data. For cross-cultural comparisons, these models take advantage of the nested nature of the data, yielding more statistical power relative to the traditional approach of calculating correlations based on aggregated nation-level data [29].

Additionally, for all analyses, we report the results from a model with all of the controls included simultaneously in the main text, and the results from a base model with no controls in the electronic supplementary material. We note the pattern of results of the models without controls in the main text.

Data for this project were collected in 2016, and the analysis plan was pre-registered in 2019, prior to the data analysis for this project. The idea for the current project came from observing the overall pattern of variation in sex differences in mate preferences across countries in a prior study using the same mate preference data [22]. To mitigate our own biases, we pre-registered our analysis plan for the current project before examining sex ratio as a possible source of variation. All data analysis was done in R. The pre-registered analysis plan, analysis script and data

can be found on the Open Science Framework: https://osf.io/fpsm6.

### (i) Relative mate preferences

Relative preferences are calculated from absolute trait preferences but incorporate the trait distribution (mean and standard deviation) of each sex in each country, like a $z$-score. The reason for including relative preferences is to account for the fact that the same absolute preferred trait value may be more or less demanding depending on the availability of that trait in the local population. Therefore, relative mate preferences were calculated using the following formula:

$$\text{relative preference} = \frac{\text{preference} - M_{\text{opp. sex trait}}}{\text{s.d.}_{\text{opp. sex trait}}}.$$

Each participant has a relative preference value for each trait, which indicates how high or low their ideal preference is for each trait, relative to the average trait level found in the opposite sex in their country. In the preregistration, relative mate preferences were originally referred to as standards. We changed the term later for clarity.

## 3. Results

### (a) Absolute mate preferences and sex ratio

Table 1 shows the results of multilevel models predicting absolute ideal mate preferences from sex and sex ratio, with control variables. The interaction between sex and sex ratio predicted absolute preference for physical attractiveness for every measure of sex ratio. Additionally, the interaction between sex and sex ratio at birth predicted most absolute mate preferences, with the exception of kindness. Effect sizes for all significant models are in the electronic supplementary materials. Removing control variables did not change the pattern of results (see electronic supplementary material).

The number of models fitted may give cause for concern about alpha inflation. However, the intention of our analyses was to reveal any overall patterns between sex ratio generally and each preference, rather than detect individual significant effects. For this reason, multiple comparison corrections may be overly conservative. Although not pre-registered, to remove any such concerns we report both unadjusted $p$-values and $p$-values adjusted for multiple comparisons using Holm–Bonferroni corrections. For the purposes of these corrections, we corrected the $p$-values associated with the interactions and we considered the test families to be all analyses using any one of the sex ratio measures (e.g. sex and sex ratio at birth predicting good financial prospects, physical attractiveness, intelligence, kindness and health). In table 1, we report which models remained significant after the correction. For adjusted $p$-values, see the electronic supplementary material.

### (i) Physical attractiveness

In general, as men became more numerous, women, relative to men, tended to increase their preference for physical attractiveness, whereas men, relative to women, decreased their preference for physical attractiveness (figure 1). The magnitude of these simple slopes varied depending on the specific measure of sex ratio used; men had significantly negative slopes for adult sex ratio and city sex ratio, whereas

women had significantly positive slopes for sex ratio ages 15–49 and sex ratio ages 15–64. All other simple slopes were not significantly different from zero (all $p$-values greater than 0.05); however, the relative differences still moved in the predicted direction. Overall, regardless of the sex ratio measure, the sex difference in absolute preference for physical attractiveness narrowed as the number of men, relative to women, increased.

### (ii) Absolute mate preferences and sex ratio at birth

The interaction between sex ratio at birth and sex additionally predicted absolute preference for physical attractiveness, good financial prospects, intelligence and health. Generally, as sex ratio at birth skewed toward female scarcity, women's preferences tended to increase while men's preferences decreased (see electronic supplementary material for figure). Specifically, simple slopes did not significantly differ from zero for physical attractiveness and health; however, the relative slopes were in the predicted direction. Additionally, while women's absolute preference for good financial prospects increased, $b = 0.09$, s.e. $= 0.04$, $p = 0.023$, men's slope did not significantly differ from zero. Lastly, men's absolute preference for intelligence significantly decreased as men became more numerous, $b = -0.14$, s.e. $= 0.03$, $p < 0.001$. Contrary to prediction, women's intelligence preferences decreased as well, $b = -0.07$, s.e. $= 0.03$, $p = 0.039$, however to a lesser degree than did men's.

### (b) Relative mate preferences and sex ratio

Table 1 shows the results of multilevel models predicting relative ideal mate preferences from sex and sex ratio, with control variables; results from models with relative preferences as the dependent variable are shown in parentheses. The interaction between sex and sex ratio predicted relative preference for good financial prospects and relative preference for physical attractiveness for every measure of sex ratio. Additionally, sex ratio at birth and sex ratio ages 15–64 predicted relative preference for health. However, because this result is not consistent across different measures of sex ratio, we do not focus on these analyses. Removing control variables did not change the overall pattern of results (see electronic supplementary material).

### (i) Good financial prospects

In general, as men became more numerous, men, compared to women, decreased their relative preferences for good financial prospects, whereas women, compared to men, tended to increase their relative preferences for good financial prospects (figure 1). All simple slopes were not significantly different from zero (all $p$-values greater than 0.05), however the relative differences still moved in the predicted direction. Overall, the sex difference in relative preference for good financial prospects widened as sex ratio increased.

### (ii) Physical attractiveness

In general, as with absolute preference, as men became more numerous, men decreased their relative preference for physical attractiveness, whereas women tended to increase their relative preference for physical attractiveness (figure 1). The magnitude of these simple slopes varied depending on the specific measure of sex ratio used; men had significantly

**Table 1.** The interaction between sex and sex ratio predicting absolute and relative mate preferences.

| preference | sex ratio measure | b (sex ratio×sex) | s.e. | p |
|---|---|---|---|---|
| good financial prospects | birth | −0.088 (−0.099) | 0.025 (0.043) | 0.001**a (0.027*) |
| | adult | −0.040 (−0.092) | 0.029 (0.043) | 0.174 (0.037*) |
| | 15–49 | −0.061 (−0.114) | 0.026 (0.037) | 0.025* (0.004**a) |
| | 15–64 | −0.048 (−0.103) | 0.028 (0.040) | 0.087 (0.013*) |
| | city | −0.044 (−0.084) | 0.027 (0.037) | 0.108 (0.028*) |
| physical attractiveness | birth | −0.095 (−0.082) | 0.025 (0.040) | <0.001***a (0.049*) |
| | adult | −0.084 (−0.118) | 0.027 (0.038) | 0.004**a (0.003**a) |
| | 15–49 | −0.115 (−0.131) | 0.023 (0.033) | <0.001***a (<0.001***a) |
| | 15–64 | −0.108 (−0.122) | 0.024 (0.035) | <0.001***a (0.001**a) |
| | city | −0.083 (−0.123) | 0.026 (0.033) | 0.002**a (<0.001***a) |
| intelligence | birth | −0.076 (−0.025) | 0.023 (0.043) | 0.002**a (0.568) |
| | adult | −0.014 (−0.011) | 0.026 (0.042) | 0.604 (0.789) |
| | 15–49 | −0.031 (−0.033) | 0.025 (0.038) | 0.212 (0.383) |
| | 15–64 | −0.018 (−0.011) | 0.025 (0.039) | 0.486 (0.784) |
| | city | −0.004 (−0.006) | 0.026 (0.039) | 0.885 (0.870) |
| kindness | birth | −0.011 (−0.002) | 0.024 (0.038) | 0.668 (0.965) |
| | adult | −0.013 (−0.021) | 0.025 (0.037) | 0.598 (0.578) |
| | 15–49 | −0.004 (−0.032) | 0.024 (0.033) | 0.856 (0.330) |
| | 15–64 | −0.016 (−0.036) | 0.024 (0.034) | 0.497 (0.295) |
| | city | −0.021 (−0.037) | 0.023 (0.034) | 0.362 (0.288) |
| health | birth | −0.085 (−0.081) | 0.023 (0.039) | <0.001***a (0.044*) |
| | adult | −0.023 (−0.048) | 0.027 (0.039) | 0.401 (0.226) |
| | 15–49 | −0.038 (−0.069) | 0.025 (0.034) | 0.134 (0.051) |
| | 15–64 | −0.034 (−0.074) | 0.025 (0.036) | 0.183 (0.045*) |
| | city | −0.021 (−0.056) | 0.024 (0.036) | 0.391 (0.123) |

*$p < 0.05$; **$p < 0.01$; ***$p < 0.001$. Results for relative mate preferences shown in parentheses.
aRemained significant after Holm–Bonferroni correction.

negative slopes for adult sex ratio and city sex ratio, whereas women had significantly positive slopes for sex ratio ages 15–49 and sex ratio ages 15–64. All other simple slopes were not significantly different from zero (all $p$-values greater than 0.05), however the relative differences still moved in the predicted direction. Overall, regardless of the sex ratio measure, the sex difference in relative preference for physical attractiveness narrowed as the number of men, relative to women, increased.

### (iii) Additional analyses

Though not in the preregistration, in order to address concerns about non-independence between countries and cities, we further examined the effect of including controls for both country and city-level proximity [56] and language [57]. Additionally, we examined the effect of nesting countries by language. Including these controls or changes in model structure did not change the pattern of results (see the electronic supplementary material).

We included a covariation summary table in the electronic supplementary material that includes the dependent variables and continuous control variables. Owing to concerns about preferences not being independent (correlations between preferences range from $r = 0.22$ to $r = 0.39$), we also conducted a principal components analysis, and used the principal components as dependent variables to test if the results had a similar pattern when preferences were no longer independent. Overall, the pattern of results remained consistent with the main analyses results (see electronic supplementary material).

### 4. Discussion

The consequences of sex ratio skew have long been of interest to scientists of evolution and behaviour, and particularly of interest to those who study mating [18,58]. Additionally, more recent work has examined the complex role of mate scarcity or abundance in patterns of sex differentiated reproductive behaviour, such as mate competition and parental care across species [59]. Despite these important advances, empirical work connecting human mate preferences to sex ratio remains scarce (for review, see [60]). Here, we attempted to address this literature gap with a large, cross-cultural investigation of human mate preferences. Overall, we found that sex differences in mate preferences vary across sex ratios. Where men are numerous, compared to where they

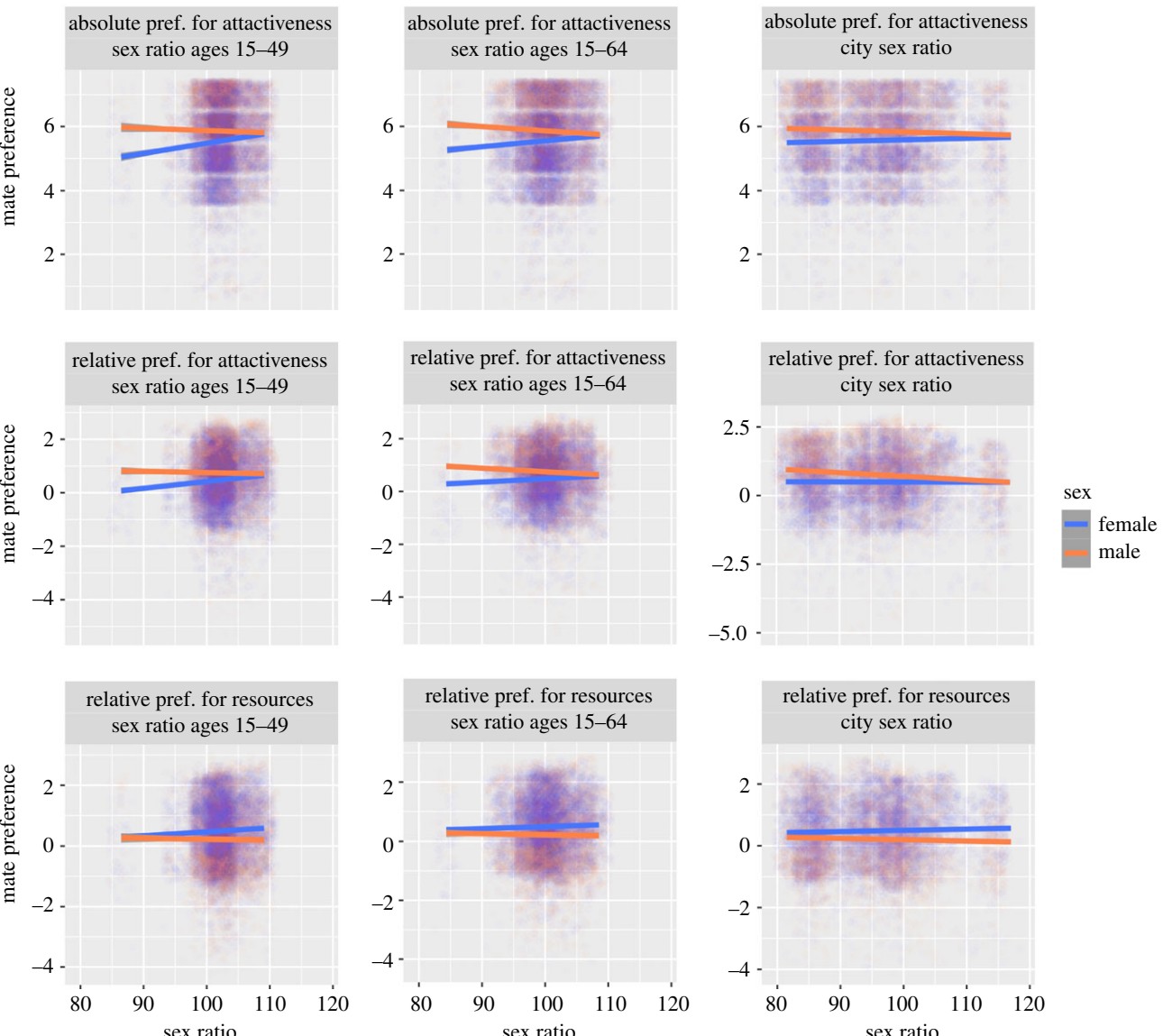

**Figure 1.** Participant mate preferences across sex ratios. Data are jittered to reduce overplotting. Regression lines, separated by sex, shown with shaded areas indicating 95% confidence intervals. The specific preference (absolute preference for physical attractiveness; relative preference for physical attractiveness; relative preference for good financial prospects) and specific sex ratio (sex ratio ages 15–49; sex ratio ages 15–64; city sex ratio) can be identified in each plot label. Sex ratio is the number of males per 100 females. (Online version in colour.)

are scarce, men tended to have lower absolute preferences for physical attractiveness, whereas women tended to have higher preferences. This inverse relationship also held for relative preferences for both physical attractiveness and good financial prospects. In sum, each sex tended to report more demanding preferences for attractiveness and resources where they had more power of choice on the mating market, compared to where they had less mating market power.

These findings are important for several reasons. First, the pattern whereby the scarcer sex sets more demanding preferences falls parsimoniously in line with patterns found for mating strategies in humans [15,17], and for mating systems, mate competition and mate preferences in non-humans [3,5,6]. While this study is correlational in nature and cannot speak to causality, the pattern of results is what would be expected if preferences for attractiveness and resources were calibrated to mate availability, and thus plastic in response to mating market demand.

Second, as we show that men's and women's preferences vary across sex ratios inversely, the magnitude of average

sex differences in preferences also varies. Much research has examined the universality of sex differences in human mate preferences [21,61]. Less research has examined the variation in sex differences across cultures. The fact that sex ratio has the power to predict cross-cultural variation in mate preferences attains special importance as two previously reported sources of variation, pathogen prevalence and gender equality, have recently failed to replicate as predictors of cross-cultural variation in human mate preferences [22,44,62,63].

Third, that sex ratio more clearly predicts variation in relative preferences than in absolute preferences has implications for the measurement and analysis of mate preference variables. While absolute preferences reflect the trait values that people desire in potential mates, they do not as directly indicate how demanding that preference is within a particular environment. For instance, a strong preference for kindness (7 on a 7-point scale) may correspond to an extremely demanding preference in an environment where the average kindness is 4 on the same scale, or a somewhat

demanding preference if the average kindness is 6 on the same scale. Given that scarcity on the mating market is hypothesized to afford power to express more stringent demands, measuring preferences in absolute terms might miss out on a critical dimension of variation relevant to sex ratio. Relative preferences, which incorporate information about the distribution of local trait values, may provide a more relevant measure of preferences because they more directly measure how demanding a given preference value is in a participant's local context.

Despite these important findings, the study does have some limitations and leaves open some important questions. First, the relationship between sex ratio and mate preferences was not as robust for some mate preference dimensions: kindness, health and intelligence. One possibility for why the same pattern did not emerge for these preferences is because they are so highly desired, and therefore more invariant. Indeed, the mean preference for kindness across all countries was, on a 7-point scale, $M = 6.23$, 95% CI [6.21, 6.26], $Mdn = 6$, for women, and $M = 6.12$, 95% CI [6.10, 6.15], $Mdn = 6$, for men. These universal near-ceiling effects leave limited room for variation. Furthermore, kindness, health and intelligence are also qualities considered very important for both men and women, and therefore these preferences may be less likely to shift downward, even when market power is low [21,64]. Future research could examine the relationship between sex ratio and a wider range of mate preferences—crucially, including those that exhibit more variation—to determine the extent of the relationship between mate preferences and sex ratio.

Second, our finding that mate preferences vary according to current sex ratio at birth could be considered somewhat surprising. Theoretically, sex ratio at birth, the number of males born for every 100 females born, does not appear to typify the conceptual variable of interest: the number of mates available to members of each sex. However, sex ratio at birth is moderately correlated with the other measures of sex ratio ($r = 0.35$, adult sex ratio; $r = 0.39$, sex ratio 15–49; $r = 0.38$, sex ratio 15–65; $r = 0.16$, city sex ratio), so it may be capturing sex ratio variation similar to adult sex ratio measures. Additionally, sex ratio at birth is an important variable to consider because it may be the origin of some skewed adult sex ratios, particularly in countries with an abundance of men. In particular, sex ratio at birth may reflect aspects of gender relations. Though skewed sex ratios can occur because of migration, violence and unbalanced death rates, sex ratio can also vary due to cultural practices such as sex selective abortions based on preferences for sons [65]. Some prior work has hypothesized that in places where women are scarce, women may have less structural power overall, and may be unable to fulfil their mate preferences even when they hold mating market power [18]. Although we did not find evidence consistent with this hypothesis—women's preferences tended to increase (not decrease) as they became scarcer—future work should continue to explore the source of sex ratio at birth's predictive power, including its potential relationship to gender equality.

Relatedly, our data do not speak to how the relationship between sex ratio and mate preferences emerges. One possibility is that the effects of sex ratio reflect evoked culture, and mating psychology reacts facultatively to local sex ratio to calibrate mate preferences. Alternatively, this relationship could reflect transmitted culture if, for example, people with less strict preferences tend to experience greater mating success when their own sex is abundant, and others mimic their preferences via prestige-based learning [66]. These possibilities are each equally consistent with the data we have here. Future research should explore further the particular ontogenetic mechanisms responsible for cross-cultural variation in preferences.

Furthermore, sex ratio measurement is made complicated by the fact that previous research has varied in the way sex ratio is defined. In particular, prior studies vary with respect to the age ranges used to estimate sex ratio, and whether operational sex ratio (only individuals able to reproduce) or adult sex ratio (all individuals considered adults, including elderly), is the key measure of sex ratio. Some of the inconsistent results in the prior literature may be due to researchers' use of only a single measure of sex ratio, which at times may fail to accurately capture the conceptual variable of interest: the availability of potential mates. Here we attempted to address this limitation by operationalizing country-level sex ratio measures in a variety of ways, and including city-level sex ratio and sex ratio at birth. By taking a broad approach to measuring sex ratio, we showed that results tended to remain robust across measures, though there were exceptions. However, a limitation of this broad approach is that it remains unclear what precisely is the best way to measure sex ratio for human mating research—a question future research must explore.

Part of the lack of clarity about how to operationalize sex ratio comes from the lack of clarity about how humans actually track mate availability. Country-level measures, or even city-level measures of sex ratio, may not accurately represent the sex ratios experienced and tracked by individual participants. More precise sex ratio measurements may produce different results than those found here.

Overall, the consequences of sex ratio have been well studied across mating behaviour in the non-human literature, from intrasexual competition, to preferences, to mating system [3,5,6]. The consequences of sex ratio have also been examined in the human literature in areas spanning from violence to financial behaviour or mating strategy [15,67,68]. However, the question of how sex ratio relates to human mate preferences has received limited attention and prior findings have lacked clarity. Here we provided evidence that sex ratio is related to mate preferences across cultures, such that where each sex is scarce, that sex tends to have higher preference demands for attractiveness and resources. These findings further elucidate the nature of human mating psychology, in particular its universal structure and systematic variation.

**Ethics.** This study was approved by the following institutional review boards and ethics committees: Ethical Committee of the Institute of Psychology, University of Wroclaw; The Survey and Behavioural Research Ethics Committee at the Chinese University of Hong Kong; The ANU Human Research Ethics Committee at The Australian National University; The Ethical Review Board of Vrije Universiteit Amsterdam; Ethics committee of the Department of Psychology, Faculty of Humanities and Social Sciences, University of Zagreb; University of Crete Psychology Department Research Ethics Committee; South-West University Neofit Rilski, Department of Psychology; Research Ethics Committee of the University of Tartu (UTREC); Ethical Committee of the Technical University of Dresden; Ethical Commission in Research of the ENES, UNAM, Morelia; Ethics Review Board of CUFE Business School; Scientific

Council of the Institute of Ethnology and Anthropology, RAS, Moscow, Russia; Ethics Council of the University of Setif 2, Algeria; Institutional Review Board of the University of Texas at Austin. Informed consent was obtained from all participants.

Data accessibility. The pre-registered analysis plan, analysis script, and data can be found on the Open Science Framework: https://osf.io/fpsm6.

Authors' contributions. K.V.W.: conceptualization, formal analysis, methodology, visualization, writing-original draft, writing-review and editing; D.C.-B.: conceptualization, investigation, methodology, visualization, writing-review and editing; D.M.B. and K.A.: conceptualization, investigation, writing-review and editing; A.S. and P.S.: investigation, project administration, writing-review and editing; T.A., G.A., M.M.A., C.A., N.A., A.A., C.S.A., D.A.D., R.A., C.B., M. Bendixen, A.B., B.B., M. Boussena, M. Butovskaya, S.C., K.C., A.C., H.C., I.C., R.M.C., M.C., D.D., S.D., I.D., B.E., A.E., I.E., C.S.E., L.F., T.F., J.C.G., K.U.G., F.G., P.G., M. Halamová, I. Herak, M. Horvat, I. Hromatko, C.H., J.L.J., F.J., K.K., T.K., L.E.O.K., N.K., T.T.K.H., I.A.K., N.K., H.M.L., A.L., G.R.L., E.L., T.L., T.T.L., G.L., N.V.L., A.M.-G., Z.M., R.M., S.L.M., N.M., G.M., C.M., E.M., A.M., B.M., J.C.N., A.N., G.N., E.O., A.O., M.S.O.-F., E.I.O., B.O., A.F.P., V.P., M.P., F.P., A.P., K.P., E.P., C.P., P.P., M.R., M.S., S. Salkičević, R.S., I.S.-S., S. Schmehl, S. Sharad, D.R.S.S., F.S.: investigation, writing-review and editing.

All authors gave final approval for publication and agreed to be held accountable for the work performed therein.

Competing interests. We declare we have no competing interests.

Funding. This material is based on work supported by the National Science Foundation under grant no. 1845586. The work of T.T.K.H. was supported by grant no. 501.01-2016.02 from the Vietnam National Foundation for Science and Technology Development (NAFOSTED). A.O. was supported by the Ministry of Science and Higher Education (grant no. 626/STYP/12/2017). A.S. and P.S. were supported by National Science Center—Poland (grant no. 2014/13/B/HS6/02644). Marina Butovskaya and D.D. were supported by State assignment project No. 01201370995 of the Institute of Ethnology and Anthropology, Moscow, Russia. P.G., A.L. and N.M. were supported by the Hungarian Scientific Research Fund—(OTKA; grant no. K125437). F.J. was supported by the National Nature Science Foundation of China (grant no. 71971225). G.A. was supported by UKRI/GCRF Gender, Justice, Security Grant (grant no. AH/S004025/1).

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
