## [Peer Review File · Proceedings of the Royal Society B: Biological Sciences]

Sex differences in human mate preferences vary across sex ratios

Kathryn V. Walter, Daniel Conroy-Beam, David M. Buss, Kelly Asao, Agnieszka Sorokowska, Piotr Sorokowski, Toivo Aavik, Grace Akello, Mohammad Madallh Alhabahba, Charlotte Alm, Naumana Amjad, Afifa Anjum, Chiemezie S. Atama, Derya Atamtürk Duyar, Richard Ayebare, Carlota Batres, Mons Bendixen, Aicha Bensafia, Boris Bizumic, Mahmoud Boussena, Marina Butovskaya, Seda Can, Katarzyna Cantarero, Antonin Carrier, Hakan Cetinkaya, Ilona Croy, Rosa María Cueto, Marcin Czub, Daria Dronova, Seda Dural, Izzet Duyar, Berna Ertugrul, Agustín Espinosa, Ignacio Estevan, Carla Sofia Esteves, Luxi Fang, Tomasz Frackowiak, Jorge Contreras Garduño, Karina Ugalde González, Farida Guemaz, Petra Gyuris, Mária Halamová, Iskra Herak, Marina Horvat, Ivana Hromatko, Chin-Ming Hui, Jas Laile Jaafar, Feng Jiang, Konstantinos Kafetsios, Tina Kavcic, Leif Edward Ottesen Kennair, Nicolas Kervyn, Truong Thi Khanh Ha, Imran Ahmed Khilji, Nils C. Köbis, Hoang Moc Lan, Andrés Láng, Georgina R. Lennard, Ernesto León, Torun Lindholm, Trinh Thi Linh, Giulia Lopez, Nguyen Van Luot, Alvaro Mailhos, Zoi Manesi, Rocio Martinez, Sarah L. McKerchar, Norbert Meskó, Girishwar Misra, Conal Monaghan, Emanuel C. Mora, Alba Moya-Garófano, Bojan Musil, Jean Carlos Natividade, Agnieszka Niemczyk, George Nizharadze, Elisabeth Oberzaucher, Anna Oleszkiewicz, Mohd Sofian Omar-Fauzee, Ike E. Onyishi, Baris Özener, Ariela Francesca Pagani, Vilmante Pakalniskiene, Miriam Parise, Farid Pazhoohi, Annette Pisanski, Katarzyna Pisanski,, Edna Ponciano, Camelia Popa, Pavol Prokop, Muhammad Rizwan, Mario Sainz, Svjetlana Salkicevic, Ruta Sargautyte, Ivan Sarmány-Schuller, Susanne Schmehl, Shivantika Sharad, Razi Sultan Siddiqui, Franco Simonetti, Stanislava Yordanova Stoyanova, Meri Tadinac, Marco Antonio Correa Varella, Christin-Melanie Vauclair, Luis Diego Vega, Dwi Ajeng Widarini, Gyesook Yoo, Marta Marta Zatková and Maja Zupancic

Article citation details

Proc. R. Soc. B **288**: 20211115.
<http://dx.doi.org/10.1098/rspb.2021.1115>

Review timeline

Original submission: 7 October 2020
1st revised submission: 17 May 2021
2nd revised submission: 28 June 2021
Final acceptance: 28 June 2021

Note: Reports are unedited and appear as submitted by the referee. The review history appears in chronological order.

Review History

RSPB-2020-2503.R0 (Original submission)

Review form: Reviewer 1

Recommendation

Accept with minor revision (please list in comments)

Scientific importance: Is the manuscript an original and important contribution to its field?

Excellent

General interest: Is the paper of sufficient general interest?

Excellent

Quality of the paper: Is the overall quality of the paper suitable?

Excellent

Is the length of the paper justified?

Yes

Should the paper be seen by a specialist statistical reviewer?

No

Do you have any concerns about statistical analyses in this paper? If so, please specify them explicitly in your report.

No

It is a condition of publication that authors make their supporting data, code and materials available - either as supplementary material or hosted in an external repository. Please rate, if applicable, the supporting data on the following criteria.

Is it accessible?

Yes

Is it clear?

Yes

Is it adequate?

Yes

Do you have any ethical concerns with this paper?

No

Comments to the Author

This manuscript reports results from a large, collaborative, international test of an interesting and important question: how do sex ratios affect human mate preferences? The results generally support the authors' predictions that members of the scarcer sex develop more demanding preferences for certain traits, particularly physical attractiveness and (for women more than men) financial resources.

This manuscript presents results from a broad, cross-cultural test of a pre-registered hypothesis, with over 14,000 participants from 45 countries. The large sample size, with all data collected in-person rather than online, ensures that the results reported here are robust and likely to be

replicable. A majority of the people sampled were members of the general community, rather than university students, providing an additional assurance that the results are robust, and not biased by sampling a specific subgroup of the study populations.

In lines 160-163, you state "individuals endowed with a mating psychology sensitive to these power differentials, able to upregulate the expression of sex-typical desires when one's sex is scarce and down regulate these desires when one's sex is numerous, would likely have had a competitive advantage individuals with desires that remained static in the face of shifting contexts." This statement is consistent with views in evolutionary psychology, that human behavior generally reflects evolved psychological adaptations. It seems at least worth considering the possibility, though, that mate preferences also reflect cultural evolution. Cultural traits potentially provide a mechanism for tracking features of the environment, such as sex ratios, producing responses that evolve rapidly to changing circumstances as a result. Given that this study does not address the underlying mechanisms, but instead focuses on the outcome, this point is not critical for this paper. Nonetheless, it seems worthwhile to consider the possibility that mate preferences reflect an interaction between evolved psychology and cultural evolution.

Collecting data from 45 countries represents an enormous undertaking, and an admirable effort to ensure cross-cultural validity of the findings. To further support the extent to which these findings generalize across human societies, it would be helpful to know some more details about the study subjects. Were they mainly from urban populations? How many were from rural, subsistence-level populations?

You state the the results did not depend on the control variables. Does this mean that none of the control variables covaried in any way with the dependent variables? It would be useful to provide a summary table with the associations, or lack thereof, between control variables and the dependent variables, at least in the supplementary data.

Review form: Reviewer 2

Recommendation

Major revision is needed (please make suggestions in comments)

Scientific importance: Is the manuscript an original and important contribution to its field?

Good

General interest: Is the paper of sufficient general interest?

Good

Quality of the paper: Is the overall quality of the paper suitable?

Acceptable

Is the length of the paper justified?

Yes

Should the paper be seen by a specialist statistical reviewer?

Yes

Do you have any concerns about statistical analyses in this paper? If so, please specify them explicitly in your report.

Yes

It is a condition of publication that authors make their supporting data, code and materials available - either as supplementary material or hosted in an external repository. Please rate, if applicable, the supporting data on the following criteria.

Is it accessible?

No

Is it clear?

N/A

Is it adequate?

N/A

Do you have any ethical concerns with this paper?

No

Comments to the Author

This MS describes a massive, multi-nation effort to test interesting hypotheses about mating market supply (via sex ratio) and preference expression. I think it could be a fascinating and landmark contribution. I do have concerns, however, which I lay out below.

The data are available online, but the file is too big to be viewed, so I can't comment on this aspect of the data plan. I am intrigued by the use of preregistration, but only 3 years after the completion of data collection. If the plan was formed only once the data came in, then please be upfront about this and about what happened in the design phase to arrive at these hypotheses and tests. At the moment, there is no way to reassure the reader that this was really a pre-reg and not a post-reg that looks a bit like a pre-registered analysis plan.

I am excited by the main findings: that preference strength is associated with the relative supply of the sex whose preferences are being measured. The idea that people express stronger (unrealized) preferences when they are of the scarce sex fits wonderfully with theory and confirms what a lot of individual-driven theories (evolutionary and economic) would predict.

The multi-level modeling approach seems to address some of the concerns that inhere to data of this kind: the dreaded ecological fallacy in particular. That said, I could not get a crisp idea of which levels of the analyses were "cleanest" from an ecological fallacy point of view. I also did not feel the authors gave a disinterested view of how well they have dealt with his problem.

The issue becomes very real when looking at the interesting finding that *birth* sex ratio is the most consistent predictor of preferences. This is the sex ratio least germane to mating market supply. The authors mirror my immediate concern on lines 426-429 of the discussion, where they mention the causes of skewed sex ratios and the undervaluation of women and girls. What if there were a group, or several groups of countries who, through shared cultural heritage and economic conditions, valued daughters less AND those countries tended to show the preference pattern of stronger female preferences and weaker male preferences (perhaps because undervalued women cannot achieve social mobility by means other than mating)? This could set up a spurious relationship between preference and sex ratio, and because of non-independence, it could drive the observed findings. This is one obvious scenario that needs direct attention. Still, the more general issue of just *how* vulnerable the analyses presented here are to this kind of non-independence problem needs to be addressed too.

I found the lead statement in the abstract (lines 112-4), introduction (142-5), and discussion (374-7) quite underwhelming for a paper of this kind of empirical sweep. There is a great deal of work on "preferences" in the broader sense used in the behavioral sciences. The question of these kinds of ev-psych measures of preference is a bit more open, but that is far from the most interesting thing about this MS. The notion that expressions of preference strength are plastic to sex ratio,

and possibly mating market demand is far more interesting and gets relatively little attention from the authors. Why do we expect mating preferences to show this kind of plasticity? What are its likely adaptive consequences? These are all much more lasting contributions, and the authors should attend to them.

In order to break out of the rather narrow confines of writing this for a particular kind of evolutionary psychologist, and to cement the contribution, this MS needs a theoretic update. Relying on Guttentag-Secord, Trivers, and Buss-Schmitt level thinking means this paper will date. Get to grips with Kokko & Jennions 2008, and Jennions & Fromhage's follow-ups, as well as Emily Stone's excellent review (cited in your pre-reg). They should be a good starting place for you to find where the real value of your amazing, hard-won data set is to be found.

One more point: I thought the justification and description of the method for the "relative preference" was inadequate. You may wish to take more time, perhaps in the supplement, laying this out. Don't bury it in the pre-reg.

Decision letter (RSPB-2020-2503.R0)

20-Nov-2020

Dear Ms Walter:

I am writing to inform you that your manuscript RSPB-2020-2503 entitled "Sex Differences in Human Mate Preferences Vary Across Sex Ratios" has, in its current form, been rejected for publication in Proceedings B.

This action has been taken on the advice of referees, who have recommended that substantial revisions are necessary. With this in mind we would be happy to consider a resubmission, provided the comments of the referees are fully addressed. However please note that this is not a provisional acceptance.

Sincerely,
Dr Sasha Dall
mailto: proceedingsb@royalsociety.org

Associate Editor

Comments to Author:

Thank you for submitting your manuscript entitled *Sex Differences in Human Mate Preferences Vary Across Sex Ratios* to Proceedings B. The manuscript was reviewed by two experts and myself. Both the reviewers were positive about this broad, cross-cultural test of how sex ratios affect human mate preferences and in particular praised the large sample sizes that are more representative of human society than past work. The authors use multilevel models to find evidence that members of the scarcer sex develop more demanding preferences for certain traits, particularly physical attractiveness and financial resources. However, both reviewers raised thoughtful and important questions that need to be addressed.

Specifically:

- 1) Reviewer 2 requested further details about the preregistration plan and why it was registered 3 years after the completion of data collection. This is important information that is necessary to assess the context of the preregistration plan.
- 2) The same reviewer raised concerns about non-independence across groups. I also agree with these concerns and it also ties in with the other reviewer's comment about the distribution of subjects across rural/urban areas. Can the authors show that this is not affecting the results? For instance, do preference traits strongly cluster by geography/cultural heritage/religion? If so, this would suggest that subject groups were not independent and this should be accounted for in the model, possibly by adding another nested level.
- 3) The other reviewer had important questions about the controls used in the multilevel model which I share. Furthermore, it is not clear to me why the main text presents the results without controls. This is the more robust test. I would recommend presenting the results of both the absolute and relative preference tests with controls in the main text.
- 4) I also had a number of questions about the statistical tests. It is not clear how much variation is explained by the model (e.g Fig 1). Can the authors conduct a goodness of fit model (or some equivalent) to estimate effect sizes? There should also be a correction for multiple testing for the results presented in Table 1. Finally, I wonder whether these preference traits can really be treated as independent traits? It seems to me that some of them might be highly correlated and so should not be analysed separately in the model. What is the strength of the correlation between these preference traits?

Reviewer(s)' Comments to Author:

Referee: 1

Comments to the Author(s)

This manuscript reports results from a large, collaborative, international test of an interesting and important question: how do sex ratios affect human mate preferences? The results generally support the authors' predictions that members of the scarcer sex develop more demanding preferences for certain traits, particularly physical attractiveness and (for women more than men) financial resources.

This manuscript presents results from a broad, cross-cultural test of a pre-registered hypothesis, with over 14,000 participants from 45 countries. The large sample size, with all data collected in-person rather than online, ensures that the results reported here are robust and likely to be replicable. A majority of the people sampled were members of the general community, rather

than university students, providing an additional assurance that the results are robust, and not biased by sampling a specific subgroup of the study populations.

In lines 160-163, you state "individuals endowed with a mating psychology sensitive to these power differentials, able to upregulate the expression of sex-typical desires when one's sex is scarce and down regulate these desires when one's sex is numerous, would likely have had a competitive advantage individuals with desires that remained static in the face of shifting contexts." This statement is consistent with views in evolutionary psychology, that human behavior generally reflects evolved psychological adaptations. It seems at least worth considering the possibility, though, that mate preferences also reflect cultural evolution. Cultural traits potentially provide a mechanism for tracking features of the environment, such as sex ratios, producing responses that evolve rapidly to changing circumstances as a result. Given that this study does not address the underlying mechanisms, but instead focuses on the outcome, this point is not critical for this paper. Nonetheless, it seems worthwhile to consider the possibility that mate preferences reflect an interaction between evolved psychology and cultural evolution.

Collecting data from 45 countries represents an enormous undertaking, and an admirable effort to ensure cross-cultural validity of the findings. To further support the extent to which these findings generalize across human societies, it would be helpful to know some more details about the study subjects. Were they mainly from urban populations? How many were from rural, subsistence-level populations?

You state the the results did not depend on the control variables. Does this mean that none of the control variables covaried in any way with the dependent variables? It would be useful to provide a summary table with the associations, or lack thereof, between control variables and the dependent variables, at least in the supplementary data.

Referee: 2

Comments to the Author(s)

This MS describes a massive, multi-nation effort to test interesting hypotheses about mating market supply (via sex ratio) and preference expression. I think it could be a fascinating and landmark contribution. I do have concerns, however, which I lay out below.

The data are available online, but the file is too big to be viewed, so I can't comment on this aspect of the data plan. I am intrigued by the use of preregistration, but only 3 years after the completion of data collection. If the plan was formed only once the data came in, then please be upfront about this and about what happened in the design phase to arrive at these hypotheses and tests. At the moment, there is no way to reassure the reader that this was really a pre-reg and not a post-reg that looks a bit like a pre-registered analysis plan.

I am excited by the main findings: that preference strength is associated with the relative supply of the sex whose preferences are being measured. The idea that people express stronger (unrealized) preferences when they are of the scarce sex fits wonderfully with theory and confirms what a lot of individual-driven theories (evolutionary and economic) would predict.

The multi-level modeling approach seems to address some of the concerns that inhere to data of this kind: the dreaded ecological fallacy in particular. That said, I could not get a crisp idea of which levels of the analyses were "cleanest" from an ecological fallacy point of view. I also did not feel the authors gave a disinterested view of how well they have dealt with his problem.

The issue becomes very real when looking at the interesting finding that *birth* sex ratio is the most consistent predictor of preferences. This is the sex ratio least germane to mating market supply. The authors mirror my immediate concern on lines 426-429 of the discussion, where they mention the causes of skewed sex ratios and the undervaluation of women and girls. What if there were a group, or several groups of countries who, through shared cultural heritage and

economic conditions, valued daughters less AND those countries tended to show the preference pattern of stronger female preferences and weaker male preferences (perhaps because undervalued women cannot achieve social mobility by means other than mating)? This could set up a spurious relationship between preference and sex ratio, and because of non-independence, it could drive the observed findings. This is one obvious scenario that needs direct attention. Still, the more general issue of just *how* vulnerable the analyses presented here are to this kind of non-independence problem needs to be addressed too.

I found the lead statement in the abstract (lines 112-4), introduction (142-5), and discussion (374-7) quite underwhelming for a paper of this kind of empirical sweep. There is a great deal of work on "preferences" in the broader sense used in the behavioral sciences. The question of these kinds of ev-psych measures of preference is a bit more open, but that is far from the most interesting thing about this MS. The notion that expressions of preference strength are plastic to sex ratio, and possibly mating market demand is far more interesting and gets relatively little attention from the authors. Why do we expect mating preferences to show this kind of plasticity? What are its likely adaptive consequences? These are all much more lasting contributions, and the authors should attend to them.

In order to break out of the rather narrow confines of writing this for a particular kind of evolutionary psychologist, and to cement the contribution, this MS needs a theoretic update. Relying on Guttentag-Secord, Trivers, and Buss-Schmitt level thinking means this paper will date. Get to grips with Kokko & Jennions 2008, and Jennions & Fromhage's follow-ups, as well as Emily Stone's excellent review (cited in your pre-reg). They should be a good starting place for you to find where the real value of your amazing, hard-won data set is to be found.

One more point: I thought the justification and description of the method for the "relative preference" was inadequate. You may wish to take more time, perhaps in the supplement, laying this out. Don't bury it in the pre-reg.

Author's Response to Decision Letter for (RSPB-2020-2503.R0)

See Appendix A.

RSPB-2021-1115.R0

Review form: Reviewer 2

Recommendation

Accept with minor revision (please list in comments)

Scientific importance: Is the manuscript an original and important contribution to its field?

Good

General interest: Is the paper of sufficient general interest?

Good

Quality of the paper: Is the overall quality of the paper suitable?

Good

Is the length of the paper justified?

Yes

Should the paper be seen by a specialist statistical reviewer?

No

Do you have any concerns about statistical analyses in this paper? If so, please specify them explicitly in your report.

No

It is a condition of publication that authors make their supporting data, code and materials available - either as supplementary material or hosted in an external repository. Please rate, if applicable, the supporting data on the following criteria.

Is it accessible?

N/A

Is it clear?

N/A

Is it adequate?

N/A

Do you have any ethical concerns with this paper?

No

Comments to the Author

The authors have done a good job of addressing the comments in the first review. the paper appears more robust and convincing. I paid more attention tot he writing this time around.

From the very first sentence of the abstract, this MS has many over-long, wrangled sentences that don't do the authors' case any favours.

“Despite the range of literature connecting sex ratio and mating behaviours in non-human animals, research examining sex ratio and human mating is limited.” – not only is the ironic use of “literature” replicated here (without irony), but the main assertion is simply not true. Human mating and sex ratios have been extensively studied, and many of the papers are cited here. What sets this MS apart from previous work is much more than a general lack of attention to the broad questions.

I find the manuscript undersells its value and the value of this kind of human evolutionary science by seeking out and overemphasising small points of novelty. Fact is that this is the most robust and general test of an important idea that has been around since at least 1982, perhaps as early as 1977.

This is, all in all, a formidable test, done with impressive rigour and appropriate statistical methods. It is far more useful and substantial than the incrementalist claims to novelty made by the authors.

Likewise, the first sentence of the introduction ... “Sex ratio.... is a key variable in mating research” fails to provide any useful information.

I found the sentence (line 2068) “Relatedly, the ontogeny of the relationship between sex ratio and mate preferences is generally left unclear here” a complex and opaque mish-mash. What does this have to do with ontogenetic development. By “ontogeny” you mean simply “how the relationship arises”, or is there some prediction that this might specifically be ontogenetic

acquisition. I rather fear that there is a tendency to over-egg the sciencey-sounding dimension of this paper, at great expemNSE to the reader and to the sense the authors want to convey.

The paragraph starting line 2076 gets tangled in a mess when talking about the "operationalization" of sex ratio, including via the so-called "operational sex ratio". The two meanings of "operational" are largely unrelated, creating deep confusion for the reader. Just try to say what you need to say simply!

I hope the authors will take to the MS with an editors eye, looking to simplify and streamline the words so that their impact is not lost. That would do this exhaustive piece of work some justice.

Decision letter (RSPB-2021-1115.R0)

21-Jun-2021

Dear Ms Walter

I am pleased to inform you that your manuscript RSPB-2021-1115 entitled "Sex Differences in Human Mate Preferences Vary Across Sex Ratios" has been accepted for publication in Proceedings B.

The referee(s) have recommended publication, but also suggest some minor revisions to your manuscript. Therefore, I invite you to respond to the referee(s)' comments and revise your manuscript. Because the schedule for publication is very tight, it is a condition of publication that you submit the revised version of your manuscript within 7 days. If you do not think you will be able to meet this date please let us know.

- 1) A text file of the manuscript (doc, txt, rtf or tex), including the references, tables (including captions) and figure captions. Please remove any tracked changes from the text before submission. PDF files are not an accepted format for the "Main Document".
- 2) A separate electronic file of each figure (tiff, EPS or print-quality PDF preferred). The format should be produced directly from original creation package, or original software format. PowerPoint files are not accepted.
- 3) Electronic supplementary material: this should be contained in a separate file and where possible, all ESM should be combined into a single file. All supplementary materials accompanying an accepted article will be treated as in their final form. They will be published

alongside the paper on the journal website and posted on the online figshare repository. Files on figshare will be made available approximately one week before the accompanying article so that the supplementary material can be attributed a unique DOI.

If you wish to submit your data to Dryad (<http://datadryad.org/>) and have not already done so you can submit your data via this link [http://datadryad.org/submit?journalID=RSPB&manu=\(Document not available\)](http://datadryad.org/submit?journalID=RSPB&manu=(Document not available)) which will take you to your unique entry in the Dryad repository. If you have already submitted your data to dryad you can make any necessary revisions to your dataset by following the above link. Please see <https://royalsociety.org/journals/ethics-policies/data-sharing-mining/> for more details.

Sincerely,
Dr Sasha Dall
mailto:proceedingsb@royalsociety.org

Associate Editor

Comments to Author:

The authors have done an excellent job thoroughly and thoughtfully addressing the concerns raised by myself and the reviewers. One reviewer has a number of small comments regarding the wording of the manuscript that the authors may wish to consider before publication.

Reviewer(s)' Comments to Author:

Referee: 2

Comments to the Author(s).

The authors have done a good job of addressing the comments in the first review. the paper appears more robust and convincing. I paid more attention to the writing this time around.

From the very first sentence of the abstract, this MS has many over-long, wrangled sentences that don't do the authors' case any favours.

“Despite the range of literature connecting sex ratio and mating behaviours in non-human animals, research examining sex ratio and human mating is limited.” – not only is the ironic use of “literature” replicated here (without irony), but the main assertion is simply not true. Human mating and sex ratios have been extensively studied, and many of the papers are cited here. What sets this MS apart from previous work is much more than a general lack of attention to the broad questions.

I find the manuscript undersells its value and the value of this kind of human evolutionary science by seeking out and overemphasising small points of novelty. Fact is that this is the most robust and general test of an important idea that has been around since at least 1982, perhaps as early as 1977.

This is, all in all, a formidable test, done with impressive rigour and appropriate statistical methods. It is far more useful and substantial than the incrementalist claims to novelty made by the authors.

Likewise, the first sentence of the introduction ... “Sex ratio... is a key variable in mating research” fails to provide any useful information.

I found the sentence (line 2068) “Relatedly, the ontogeny of the relationship between sex ratio and mate preferences is generally left unclear here” a complex and opaque mish-mash. What does this have to do with ontogenetic development. By “ontogeny” you mean simply “how the relationship arises”, or is there some prediction that this might specifically be ontogenetic acquisition. I rather fear that there is a tendency to over-egg the sciencey-sounding dimension of this paper, at great expense to the reader and to the sense the authors want to convey.

The paragraph starting line 2076 gets tangled in a mess when talking about the “operationalization” of sex ratio, including via the so-called “operational sex ratio”. The two meanings of “operational” are largely unrelated, creating deep confusion for the reader. Just try to say what you need to say simply!

I hope the authors will take to the MS with an editor's eye, looking to simplify and streamline the words so that their impact is not lost. That would do this exhaustive piece of work some justice.

Author's Response to Decision Letter for (RSPB-2021-1115.R0)

See Appendix B.

Decision letter (RSPB-2021-1115.R1)

28-Jun-2021

Dear Ms Walter

I am pleased to inform you that your manuscript entitled "Sex Differences in Human Mate Preferences Vary Across Sex Ratios" has been accepted for publication in Proceedings B.

Data Accessibility section

Open Access

Paper charges

Sincerely,

Appendix A

May 17, 2021

Re: Resubmission of manuscript “Sex Differences in Human Mate Preferences Vary Across Sex Ratios” [RSPB-2020-2503] to Proceedings of the Royal Society B.

Dear Dr. Sasha Dall,

Thank you for the opportunity to revise and resubmit our manuscript, “Sex Differences in Human Mate Preferences Vary Across Sex Ratios”. We appreciate the thoughtful reviews and constructive suggestions provided by yourself and the reviewers. We have now revised the manuscript based on these comments and believe that the manuscript is substantially improved.

In particular, we agree with your concerns about non-independence between groups and the overall detail and rigor of the analyses. We have now done a number of additional analyses, which all point towards the robustness of our findings and add additional important information. Namely, we have run additional controls to account for shared culture between groups, computed effect sizes, corrected for multiple tests, examined the relationship between controls and the dependent variables, and re-analyzed the data using the principal components of preferences as the outcome variables rather than manifest preferences alone. We have also clarified the details of our pre-registration, our use of relative preferences, and the contributions of our manuscript.

Following this letter are your comments and the reviewer comments with our responses in italics, including how and where the text was modified. Thank you for your consideration of our revised manuscript.

Sincerely,
Kathryn V. Walter
Psychological & Brain Sciences
University of California, Santa Barbara
kwalter@ucsb.edu

Response to Editor and Reviewers

Associate Editor

Comments to Author:

Thank you for submitting your manuscript entitled Sex Differences in Human Mate Preferences Vary Across Sex Ratios to Proceedings B. The manuscript was reviewed by two experts and myself. Both the reviewers were positive about this broad, cross-cultural test of how sex ratios affect human mate preferences and in particular praised the large sample sizes that are more representative of human society than past work. The authors use multilevel models to find evidence that members of the scarcer sex develop more demanding preferences for certain traits, particularly physical attractiveness and financial resources. However, both reviewers raised thoughtful and important questions that need to be addressed.

Specifically:

1) Reviewer 2 requested further details about the preregistration plan and why it was registered 3 years after the completion of data collection. This is important information that is necessary to assess the context of the preregistration plan.

Of course, and we apologize for not being clear. The data were collected in 2016 before the idea to test mate preferences across sex ratios was conceptualized, however it is important to emphasize that we pre-registered the analysis plan for this project before running analyses to test the predictions.

We completed the analyses for another study examining sex differences in human mate preferences and variation in mate preferences due to pathogen prevalence and gender equality in late 2018 (Walter et al., 2020). Then after observing the pattern of the data from that previous study, we noticed that China (known to have a skewed sex ratio) showed a reversal of the pattern of sex difference for physical attractiveness preferences (women preferred physical attractiveness more than men in China). As part of our research program goals are to discover sources of variation in preferences, we started thinking about sex ratio as a possible source of variation. Prior literature showed inconsistent results when examining the relationship between preferences and sex ratio, but we thought some of these inconsistencies might be due to methodology issues.

To mitigate our own biases, we pre-registered our analysis plan for our sex ratio project before running analyses to examine if sex ratio was a source of variation. This resulted in our analysis plan being pre-registered in early 2019, before we ran the analyses for this project.

We have revised the methods section to make this timeline and process clearer on lines 290-295.

2) The same reviewer raised concerns about non-independence across groups. I also agree with these concerns and it also ties in with the other reviewer's comment about the distribution of subjects across rural/urban areas. Can the authors show that this is

not affecting the results? For instance, do preference traits strongly cluster by geography/cultural heritage/religion? If so, this would suggest that subject groups were not independent and this should be accounted for in the model, possibly by adding another nested level.

Absolutely, non-independence across groups is an important factor to consider. We do control for latitude, continent, and religion in our analyses. These controls do not impact the pattern of results. Additionally, the multi-level models, which nest participants by country or city, account for the fact that participants in the same country or city are likely to be more similar than those between countries and cities.

However, we have further examined the potential impact of non-independence in our data. As you mention, some countries that are clustered together may be more likely to have similar geography (e.g. through shared geographical features) and similar cultural heritage (e.g. through a history of trade). Therefore, countries or cities that share these features may not be considered independent. To try to account for this, we controlled for the most common language spoken in each country or city, because shared language may indicate a shared cultural heritage. We found that including language as a control did not change the pattern of results.

In addition, we examined the effect of nesting countries by language. We fit a separate series of multilevel models in which participants were nested within countries and countries were nested within national languages, with random slopes for the sex effect at the language level. These models were an attempt to address patterns of dependency reflected in shared language (e.g. shared cultural history). In these models, the pattern of results was consistent with those found in the main text. If anything, the relationship between sex ratio and relative preferences was stronger in this set of analyses, with more models showing significant relationships between sex ratio and the interaction between sex and relative preferences when using common language as a grouping variable.

We also examined the effect of proximity on our pattern of results by accounting for spatial correlations among countries and cities. Due to the complexity of these analyses, we were only able to examine the role of proximity in simplified versions of our main models. We found however, that these simplified models approximated our main results well and that controlling for proximity did not change the pattern of results.

The results of these models are fully reported and discussed in the supplemental material on lines 384-449. They are also summarized in the main text on lines 388-393.

3) The other reviewer had important questions about the controls used in the multilevel model which I share. Furthermore, it is not clear to me why the main text presents the results without controls. This is the more robust test. I would recommend presenting the results of both the absolute and relative preference tests with controls in the main text.

Thank you for the suggestion. We just presented the simplest model in the main text, and as adding in the controls did not change the pattern of results, we reported the results of those models in the supplement. However, we are happy to switch the two and include the models with

controls in the main text and the models without controls in the supplemental material. You will find that now the models with controls are in the main text on lines 329-332, and the models without controls are in the supplementary material on lines 240-259.

4) I also had a number of questions about the statistical tests. It is not clear how much variation is explained by the model (e.g Fig 1). Can the authors conduct a goodness of fit model (or some equivalent) to estimate effect sizes? There should also be a correction for multiple testing for the results presented in Table 1. Finally, I wonder whether these preference traits can really be treated as independent traits? It seems to me that some of them might be highly correlated and so should not be analysed separately in the model. What is the strength of the correlation between these preference traits?

Absolutely. Computing effect sizes for multilevel models can be complicated. However, we agree this is important information. To provide this, we computed effect sizes using an approximated r^2 based on the correlation between observed and model-predicted values. The effect sizes for all significant models can be found in the supplementary material on lines 357-361.

As far as correcting for multiple comparisons: we originally did not correct for multiple comparisons because our aims did not include looking for individual significant effects in this study (e.g. we were not examining which specific sex ratio measure would be significantly related to a particular preference). Rather we were looking for an overall pattern between sex ratio generally and each preference. Therefore, a correction for multiple testing may be overly conservative. However, we used the Holm-Bonferroni method and calculated adjusted p-values for the main analyses. These corrected p-values can be found in the supplementary material on lines 325-344. In the main text, we noted via a symbol, which p-values remained significant after the Holm-Bonferroni correction, found on lines 329-332. We discuss the correction in the main text on lines 318-328 and leave it to the reader to determine which is most appropriate to interpret. However, we note that the central findings persist even after correction: both absolute and relative physical attractiveness preferences are robustly predicted by sex ratio and relative preferences for good financial prospects are still significantly predicted by at least one sex ratio measure (sex ratio for ages 15-49).

Regarding the independence of preferences: conceptually speaking, the preference traits are independent. Preferences do tend to be correlated though, such that those who have higher preferences for intelligence have higher preferences for health, good financial prospects, physical attractiveness, and kindness. In this dataset, the correlations between preferences range from $r=0.22$ to $r = 0.39$. We have included a table of these correlations in the supplementary material on lines 346-355. To address this non-independence of preferences, we also conducted a principal components analysis on preferences, extracting 2 components to see if the results had a similar pattern as when preferences were treated as independent. Overall, the pattern of results remained consistent with the main analyses. These results are reported in the supplementary material on lines 362-382 and discussed in the main text on line 394-400.

Reviewer(s)' Comments to Author:

Referee: 1

Comments to the Author(s)

This manuscript reports results from a large, collaborative, international test of an interesting and important question: how do sex ratios affect human mate preferences? The results general support the authors' predictions that members of the scarcer sex develop more demanding preferences for certain traits, particularly physical attractiveness and (for women more than men) financial resources.

This manuscript presents results from a broad, cross-cultural test of a pre-registered hypothesis, with over 14,000 participants from 45 countries. The large sample size, with all data collected in-person rather than online, ensures that the results reported here are robust and likely to be replicable. A majority of the people sampled were members of the general community, rather than university students, providing an additional assurance that the results are robust, and not biased by sampling a specific subgroup of the study populations.

In lines 160-163, you state "individuals endowed with a mating psychology sensitive to these power differentials, able to upregulate the expression of sex-typical desires when one's sex is scarce and down regulate these desires when one's sex is numerous, would likely have had a competitive advantage individuals with desires that remained static in the face of shifting contexts." This statement is consistent with views in evolutionary psychology, that human behavior generally reflects evolved psychological adaptations. It seems at least worth considering the possibility, though, that mate preferences also reflect cultural evolution. Cultural traits potentially provide a mechanism for tracking features of the environment, such as sex ratios, producing responses that evolve rapidly to changing circumstances as a result. Given that this study does not address the underlying mechanisms, but instead focuses on the outcome, this point is not critical for this paper. Nonetheless, it seems worthwhile to consider the possibility that mate preferences reflect an interaction between evolved psychology and cultural evolution.

Thank you for your comments. Yes, it is true that we are not directly addressing the underlying mechanisms in this study. Therefore, it is certainly possible that the outcome is the result of another process such as an interaction between psychology and cultural evolution. We have added a paragraph which address this possibility in the discussion on lines 472-479.

Collecting data from 45 countries represents an enormous undertaking, and an admirable effort to ensure cross-cultural validity of the findings. To further support the extent to which these findings generalize across human societies, it would be helpful to know some more details about the study subjects. Were they mainly from urban populations? How many were from rural, subsistence-level populations?

Thank you for your kind comments. While the sample has multinational breadth, participants tended to be from large cities, to be well-educated, and have average economic situations. To provide more information about the study participants and to show the variation in the sample,

we have created a table in the supplementary material with the list of cities in which the data were collected in and each city's population size and density on lines 121-136. We had also collected data about the education level and economic situation of the participants and have reported this demographic information for each city in a table in the supplementary material on lines 137-148. We have briefly summarized this additional information about the populations studied in the methods section of the main text on lines 233-235.

You state the the results did not depend on the control variables. Does this mean that none of the control variables covaried in any way with the dependent variables? It would be useful to provide a summary table with the associations, or lack thereof, between control variables and the dependent variables, at least in the supplementary data.

We have created a covariation summary table of the controls and dependent variables and added it to the supplementary material on lines 346-355. The relationship between the dependent variables and the control variables was weak overall.

Referee: 2

Comments to the Author(s)

This MS describes a massive, multi-nation effort to test interesting hypotheses about mating market supply (via sex ratio) and preference expression. I think it could be a fascinating and landmark contribution. I do have concerns, however, which I lay out below.

The data are available online, but the file is too big to be viewed, so I can't comment on this aspect of the data plan. I am intrigued by the use of preregistration, but only 3 years after the completion of data collection. If the plan was formed only once the data came in, then please be upfront about this and about what happened in the design phase to arrive at these hypotheses and tests. At the moment, there is no way to reassure the reader that this was really a pre-reg and not a post-reg that looks a bit like a pre-registered analysis plan.

We are sorry to hear that the data file is too large to be viewed. Yes, the data file is quite large. To help address this we have uploaded a zipped file to the OSF page, which may make it easier to download.

Also, as we explained above (in our response to the Editor's comments), the analysis plan was indeed pre-registered before we ran the analyses for this project. The data were analyzed earlier for other projects and to test other hypotheses that did not involve sex ratio. We have clarified the timeline in the main text on lines 290-295.

I am excited by the main findings: that preference strength is associated with the relative supply of the sex whose preferences are being measured. The idea that people express stronger (unrealized) preferences when they are of the scarce sex fits wonderfully with theory and confirms what a lot of individual-driven theories (evolutionary and economic) would predict.

Thank you - we are also excited.

The multi-level modeling approach seems to address some of the concerns that inhere to data of this kind: the dreaded ecological fallacy in particular. That said, I could not get a crisp idea of which levels of the analyses were "cleanest" from an ecological fallacy point of view. I also did not feel the authors gave a disinterested view of how well they have dealt with his problem.

The issue becomes very real when looking at the interesting finding that *birth* sex ratio is the most consistent predictor of preferences. This is the sex ratio least germane to mating market supply. The authors mirror my immediate concern on lines 426-429 of the discussion, where they mention the causes of skewed sex ratios and the undervaluation of women and girls. What if there were a group, or several groups of countries who, through shared cultural heritage and economic conditions, valued daughters less AND those countries tended to show the preference pattern of stronger female preferences and weaker male preferences (perhaps because undervalued women cannot achieve social mobility by means other than mating)? This could set up a spurious relationship between preference and sex ratio, and because of non-independence, it could drive the observed findings. This is one obvious scenario that needs direct attention. Still, the more general issue of just *how* vulnerable the analyses presented here are to this kind of non-independence problem needs to be addressed too.

Yes, it is difficult to interpret the sex ratio at birth results, although when we controlled for gender equality, sex ratio at birth's effect remained the same. We agree that addressing non-independence is very important. We controlled for factors like region, religion, and GDP, but it is of course difficult to anticipate and precisely quantify all dimensions of non-independence. As we mentioned above, in these revisions we have tried to address the issue of non-independence more directly by including geographical proximity and national language as controls. These variables likely encompass information such as shared trade history and shared geographical features which may be related to shared culture. Including these additional controls did not change the pattern of results. The results are reported and discussed in the supplementary material on lines 384-449 and summarized in the main text on lines 388-393.

Additionally, the issue of causality is always challenging with this kind of approach. However, we've been careful to state that these results are consistent with sex ratio affecting mate preferences. We would not claim to have unique evidence of a causal role of sex ratio per se on mate preferences. As mentioned above, we have also added a paragraph to the discussion on lines 472-479 about the possibility of cultural evolution as the ontogenetic mechanism responsible for this pattern of cross-cultural variation in preferences.

I found the lead statement in the abstract (lines 112-4), introduction (142-5), and discussion (374-7) quite underwhelming for a paper of this kind of empirical sweep. There is a great deal of work on "preferences" in the broader sense used in the behavioral sciences. The question of these kinds of ev-psych measures of preference is

a bit more open, but that is far from the most interesting thing about this MS. The notion that expressions of preference strength are plastic to sex ratio, and possibly mating market demand is far more interesting and gets relatively little attention from the authors. Why do we expect mating preferences to show this kind of plasticity? What are its likely adaptive consequences? These are all much more lasting contributions, and the authors should attend to them.

We appreciate that you think that the paper could make a lasting contribution. However, we want to be careful, as we only have correlational data, to not over-interpret the results. Theoretically, the hypothesis is that preference demands calibrate to an individual's circumstances. Throughout human evolutionary history, those individuals with flexible preferences would likely have had relative reproductive success compared to those who had preferences static to changing circumstances (this logic is stated in lines 164-168). Here, we examine specifically the circumstance of power on the mating market via sex ratio as a potential cue for preference calibration. We find evidence that preferences for attractiveness and resources do vary across the abundance or scarcity of an individual's own sex, on average. However, with the current data, we cannot confidently claim that mate preferences are plastic or infer that sex ratio causes changes in preferences. We would need a different, longitudinal dataset to address those specific claims. However, the current evidence is what we would expect to find, if what you suggest, that preference demands are plastic to sex ratio, is in fact the case. Though we would like to be cautious to not overextend the inferences we can make from this particular dataset, and therefore we make modest claims that are supported by the data: that preference demand varies across sex ratios. We agree that it is very interesting that this may suggest that preferences are flexible to circumstances in predictable ways.

We have included a statement in the discussion, lines 419-421, about this possibility.

In order to break out of the rather narrow confines of writing this for a particular kind of evolutionary psychologist, and to cement the contribution, this MS needs a theoretic update. Relying on Guttentag-Secord, Trivers, and Buss-Schmitt level thinking means this paper will date. Get to grips with Kokko & Jennions 2008, and Jennions & Fromhage's follow-ups, as well as Emily Stone's excellent review (cited in your pre-reg). They should be a good starting place for you to find where the real value of your amazing, hard-won data set is to be found.

We agree that the papers you mention (Kokko & Jennions, 2008; Jennions & Fromhage, 2017; Stone, 2018) are important. We have mentioned this more recent work in the discussion on lines 403-407.

One more point: I thought the justification and description of the method for the "relative preference" was inadequate. You may wish to take more time, perhaps in the supplement, laying this out. Don't bury it in the pre-reg.

Thank you for bringing this to our attention. We have included more justification and a more detailed description of relative preferences in the methods section on lines 300-307.

Appendix B

June 27, 2021

Re: Resubmission of manuscript “Sex Differences in Human Mate Preferences Vary Across Sex Ratios” [RSPB-2021-1115] to Proceedings of the Royal Society B.

Dear Dr. Sasha Dall,

We are very excited that our manuscript, “Sex Differences in Human Mate Preferences Vary Across Sex Ratios,” has been accepted for publication with minor revisions. We appreciate the constructive wording suggestions provided by the reviewer. We have revised the manuscript based on these comments and believe that the manuscript is much improved. In particular, we have broken up long sentences, clarified statements, and simplified some of the language. The manuscript is now easier to understand and its impact is clearer.

Following this letter are your comments and the reviewer comments with our responses in italics, including how and where the text was modified. Following that is a tracked changes version of the revised manuscript. Thank you for your consideration of our revised manuscript.

Sincerely,
Kathryn V. Walter
Psychological & Brain Sciences
University of California, Santa Barbara
kwalter@ucsb.edu

Associate Editor

Comments to Author:

The authors have done an excellent job thoroughly and thoughtfully addressing the concerns raised by myself and the reviewers. One reviewer has a number of small comments regarding the wording of the manuscript that the authors may wish to consider before publication.

Thank you for your kind comments. We have addressed the wording issues raised by the reviewer and believe that the manuscript is easier to understand and its impact is now clearer.

Reviewer(s)' Comments to Author:

Referee: 2

Comments to the Author(s).

The authors have done a good job of addressing the comments in the first review. The paper appears more robust and convincing. I paid more attention to the writing this time around.

From the very first sentence of the abstract, this MS has many over-long, wrangled sentences that don't do the authors' case any favours.

“Despite the range of literature connecting sex ratio and mating behaviours in non-human animals, research examining sex ratio and human mating is limited.” – not only is the ironic use of “literature” replicated here (without irony), but the main assertion is simply not true. Human mating and sex ratios have been extensively studied, and many of the papers are cited here. What sets this MS apart from previous work is much more than a general lack of attention to the broad questions.

We agree that human mating and sex ratio have been studied extensively. However, the mating topics that have been examined are limited in scope, in that researchers tend to focus on a particular set of variables, namely, mating strategies and mating behaviors. The relationship between mate preferences and sex ratio has been given far less attention empirically, though prior theoretical work has made predictions about this relationship. In the abstract, lines 118-121, we have split the first sentence into two and have clarified what we mean by limited. In doing so, we have more precisely oriented our manuscript in relation to the prior literature.

I find the manuscript undersells its value and the value of this kind of human evolutionary science by seeking out and overemphasising small points of novelty. Fact is that this is the most robust and general test of an important idea that has been around since at least 1982, perhaps as early as 1977.

This is, all in all, a formidable test, done with impressive rigour and appropriate statistical methods. It is far more useful and substantial than the incrementalist claims to novelty made by the authors.

We appreciate that you see the value in our manuscript. We agree that a strength of our manuscript is our ability to rigorously test variation in mate preferences across sex ratios. We do think that this work fills a gap in the literature by providing a strong empirical test of the relationship between sex ratio and mate preferences and we think that this is shown throughout the manuscript.

Likewise, the first sentence of the introduction ... “Sex ratio.... is a key variable in mating research” fails to provide any useful information.

Thank you for pointing this out, we have now removed that part of the sentence. The first sentence is now improved on lines 137-138.

I found the sentence (line 2068) “Relatedly, the ontogeny of the relationship between sex ratio and mate preferences is generally left unclear here” a complex and opaque mish-mash. What does this have to do with ontogenetic development. By “ontogeny” you mean simply “how the relationship arises”, or is there some prediction that this might specifically be ontogenetic acquisition. I rather fear that there is a tendency to over-egg the sciencey-sounding dimension of this paper, at great expense to the reader and to the sense the authors want to convey.

We agree that we want to make the paper easy for readers to understand. We have revised this sentence to clarify that we are discussing possible ways for this relationship to develop, on lines 473-474.

The paragraph starting line 2076 gets tangled in a mess when talking about the “operationalization” of sex ratio, including via the so-called “operational sex ratio”. The two meanings of “operational” are largely unrelated, creating deep confusion for the reader. Just try to say what you need to say simply!

We appreciate that the different versions of the word “operational” may be somewhat confusing. We have revised the paragraph to eliminate this confusion, by using different words to describe operationalization on lines 481-493. We have also fixed a similar issue in the introduction on lines 201-205.

I hope the authors will take to the MS with an editors eye, looking to simplify and streamline the words so that their impact is not lost. That would do this exhaustive piece of work some justice.

We have carefully gone through the manuscript and separated really long sentences into two as well as simplified the language as much as possible.